# Performance of Convolutional Neural Networks for Polyp Localization on Public Colonoscopy Image Datasets

**DOI:** 10.3390/diagnostics12040898

**Published:** 2022-04-04

**Authors:** Alba Nogueira-Rodríguez, Miguel Reboiro-Jato, Daniel Glez-Peña, Hugo López-Fernández

**Affiliations:** 1CINBIO, Department of Computer Science, ESEI-Escuela Superior de Ingeniería Informática, Universidade de Vigo, 32004 Ourense, Spain; alnogueira@uvigo.es (A.N.-R.); mrjato@uvigo.es (M.R.-J.); dgpena@uvigo.es (D.G.-P.); 2SING Research Group, Galicia Sur Health Research Institute (IIS Galicia Sur), SERGAS-UVIGO, 36213 Vigo, Spain

**Keywords:** colorectal cancer, deep learning, convolutional neural network (CNN), polyp detection, polyp localization

## Abstract

Colorectal cancer is one of the most frequent malignancies. Colonoscopy is the de facto standard for precancerous lesion detection in the colon, i.e., polyps, during screening studies or after facultative recommendation. In recent years, artificial intelligence, and especially deep learning techniques such as convolutional neural networks, have been applied to polyp detection and localization in order to develop real-time CADe systems. However, the performance of machine learning models is very sensitive to changes in the nature of the testing instances, especially when trying to reproduce results for totally different datasets to those used for model development, i.e., inter-dataset testing. Here, we report the results of testing of our previously published polyp detection model using ten public colonoscopy image datasets and analyze them in the context of the results of other 20 state-of-the-art publications using the same datasets. The F1-score of our recently published model was 0.88 when evaluated on a private test partition, i.e., intra-dataset testing, but it decayed, on average, by 13.65% when tested on ten public datasets. In the published research, the average intra-dataset F1-score is 0.91, and we observed that it also decays in the inter-dataset setting to an average F1-score of 0.83.

## 1. Introduction

In the last few years, significant research has been published on the application of deep learning (DL) for colorectal polyp detection and characterization in colonoscopy images, as demonstrated by the growing number of reviews on the topic [1,2,3,4]. Polyp detection is way more advanced than characterization, and several randomized control trials (RCT) have already been conducted [5,6,7,8,9,10], some of which are associated with the development of commercial systems [3].

This difference is also reflected in the availability of public colonoscopy image datasets. In the case of polyp detection, one of the most relevant events in this field was the celebration of the MICCAI 2015 conference [11], since it hosted a sub-challenge on automatic polyp detection for which the first and most well-known public colonoscopy datasets were published. These are the CVC-ClinicDB [12], ETIS-Larib [13], and ASU-Mayo Clinic Colonoscopy Video [14] datasets. Since then, several new datasets have been released, significantly increasing the number publicly available data. In addition, in the particular case of the CVC-ClinicDB dataset, its creators have extended it with three more public datasets: CVC-ColonDB [15,16], CVC-PolypHD [15,16], and CVC-ClinicVideoDB [17,18]. The growth in the volume of public data has seen a remarkable increase in recent years with the release of PICCOLO [19], Kvasir-SEG [20], LDPolypVideo [21], SUN [22], and KUMC [23], each including several thousands of polyp images, with the latter three exceeding the total volume of images published so far. All these datasets include annotations of the polyp locations as either bounding boxes or binary masks and, therefore, are suitable for polyp localization. In contrast, there are only three public datasets suitable for polyp characterization. The first dataset was published by Mesejo et al. [24] in 2016 and, since then, only the PICCOLO [19] and KUMC [23] datasets have included the necessary annotations for this task.

In our previous review [1], we collected the most relevant studies applying DL for polyp detection and characterization in colonoscopy and analyzed them from a technical point of view, focusing on the low-level details for the implementation of the DL models. Together with the review, we created a GitHub repository (https://github.com/sing-group/deep-learning-colonoscopy (accessed on 22 February 2022)) containing the most relevant information, especially the performance metrics reported by each study and the test datasets used. Since then, we have been continuously updating the repository to add new works and datasets as they were published. As a result of carrying out the work presented here, we improved the repository by adding information about the train datasets used in each analysis and by detailing the type of evaluation carried out in polyp detection. It is important to note that we do not consider preprints for inclusion in the repository due to the high activity in the field and the fact that we are not able to track and curate all of them in an appropriate and sustainable way. Nevertheless, there are some recent preprints, such as the work of Ali et al., 2021, presenting the PolypGen [25] dataset, that will be included as soon they are published in a peer-reviewed journal.

Despite this, comparing the models developed in different works is not straightforward, since they use different ways of assessing their performance. Most of them use private datasets for model development and testing, hindering the reproducibility. Other works use only public datasets for both model development and testing. Finally, there are hybrid works where the performance of a model developed with a private dataset is evaluated on a public dataset.

In a recent work, we published a real-time polyp detection model based on a YOLOv3 [26] pretrained with PASCAL VOC datasets [27] that we fine-tuned using a private dataset (28,576 annotated images from 941 different polyps). This model achieved an F1-score of 0.88 (recall = 0.87, precision = 0.89) in a bounding-box-based evaluation using still images (a test partition of our private dataset).

Nevertheless, the performance of ML models is very sensitive to changes in the nature of the testing instances when compared to those instances used for developing them, especially when trying to reproduce results on completely different datasets to those used for model development (inter-dataset testing). Aiming at gaining insights for taking further steps for improving our model following a data-centric approach, in this work, we systematically evaluate the performance of our published model, without retrain, on ten public datasets. This is, to the best of our knowledge, the first time that such extensive evaluation has been carried out. In addition, we also include a comparison of published research on polyp localization, including the best performances reported by each study on public datasets. In this regard, there is interest in comparing intra-dataset performances (i.e., a performance evaluation on a test split of the dataset used for model development, either private or public) versus inter-dataset performances (i.e., a performance evaluation on a dataset different than the one used for model development).

## 2. Materials and Methods

### 2.1. Our Polyp Localization Network

In a previous work [26], we reported the results of training and evaluating a real-time automatic polyp detection system based on YOLOv3. For this purpose, in the context of the PolyDeep project (http://polydeep.org (accessed on 22 February 2022)), we created a private dataset containing 28,576 polyp images from 941 different polyps, out of which 21 046 were acquired under white light (WL) and 7530 under narrow-band imaging (NBI) light. The images were manually annotated by expert endoscopists to specify the polyp locations as bounding boxes. This image dataset is part of a larger collection of annotated polyp videos and images, named PIBAdb, which is already available through the IISGS BioBank (https://www.iisgaliciasur.es/home/biobanco/cohorte-pibadb (accessed on 22 February 2022)).

For model development, we set aside a test partition containing 30% of the polyps (283; 8658 images) to perform a bounding-box-based evaluation. The remaining 70% was, in turn, split into train (70%; 460 polyps; 13,873 images) and validation (30%; 198 polyps; 6045 images) partitions. It is important to note that this image dataset only includes polyp images with exactly one polyp.

The YOLOv3 model used as a basis was the Apache MxNet [28] implementation that the GluonCV toolkit [29] provides pre-trained with the PASCAL VOC 2007 and 2012 challenges’ [27] train and validation datasets. This base model was fine-tuned using the train partition of our dataset, achieving an F1-score of 0.88 (recall = 0.87, precision = 0.89) and an average precision (AP) of 0.87 in a bounding-box-based evaluation using the test partition. The results are on par with other state-of-the-art models, and the model was able to process frames at a rate of 0.041 s/frame, thus being able to operate in real time.

This model was used, without retrain, to carry out the experiments described in Section 2.4 in order to evaluate its performance on different public datasets. Although the dataset used to develop the model does not include images with multiple polyps or images without polyps, given the nature of YOLOv3, the model is able to predict multiple polyps when necessary, as it will be shown.

### 2.2. Public Colonoscopy Image Datasets and Polyp Localization Studies Selection

Figure 1 shows the criteria for selecting public colonoscopy image datasets and polyp localization studies. This selection started with the 44 studies and 13 public datasets collected as of February 2022 in the GitHub repository associated with our review on DL for polyp detection and classification in colonoscopy, mentioned above.

Since one of our objectives is to draw a comparison of published research that reports performance metrics of public datasets, 17 studies were excluded in the first place because they only evaluated the models on private datasets. In addition, two datasets were also excluded for different reasons: (*i*) the CP-CHILD dataset [30] was also excluded since it only provides frames labeled as “polyp” and “non-polyp” and not a suitable ground truth including the polyp localizations; and (*ii*) the ASU-Mayo Clinic Colonoscopy Video dataset [14] was excluded since we were not able to access the dataset after repeated attempts to contact the authors without obtaining a response. Because of the exclusion of the ASU-Mayo Clinic Colonoscopy dataset, three studies that used it were also discarded.

From the remaining 23 studies, the following three were excluded: (*i*) the study by Misawa et al., 2021 [22] was excluded since they evaluate the detection performance instead of the localization performance, despite the fact they were using an object detection network architecture (YOLOv3); and (*ii*) the studies from Tashk et al., 2019 [31] and Sánchez-Peralta et al., 2020 [19] were excluded since they performed polyp segmentation and, therefore, provide pixel-based performance metrics, which are not comparable with the bounding-box-based performance metrics of the polyp localization studies. This latter cause of exclusion also motivated the discard of the CVC-EndoSceneStill dataset.

So, after applying the selection criteria seen in Figure 1, 20 studies and 10 public datasets were selected for evaluating the performance of our polyp localization model.

### 2.3. Public Colonoscopy Image Datasets Description and Preprocessing

Table 1 shows the most relevant details of the ten public colonoscopy image datasets selected for the analysis. Regarding the type of ground truth provided, seven of them provide polyps annotated with binary masks, namely CVC-ClinicDB [12], CVC-ColonDB [15,16], CVC-PolypHD [15,16], ETIS-Larib [13], CVC-ClinicVideoDB [17,18], and PICCOLO [19]. In these cases, we converted the binary masks into bounding boxes to be able to analyze them with our model (the procedure is described below in this section). Three of them provide polyp locations as bounding boxes, namely the KUMC [23], SUN [22], and LDPolypVideo [21] datasets. Finally, Kvasir-SEG [20] provides both segmentation and localization information.

The public datasets show a lot of variability in terms of number of images, number of polyps, image resolution, capturing device, etc., as shown in Table 1. The trend in the most recent datasets is to include non-polyp images. In addition, as can be seen in Figure 2, which shows one random image of each dataset included, the variability in the appearance of the images themselves and the polyps contained in them is also high (e.g., Kvasir-SEG contains images with superimposed text and/or the presence of instruments, etc.).

As explained before, almost all datasets provide polyp locations as binary masks, and thus are suitable for object segmentation models. Since our model works with bounding boxes information, the binary masks were converted into this representation using the scikit-image Python library. Figure 3 shows an example of this conversion procedure, where it can be seen that the scikit-image functions allow obtaining the minimum bounding boxes that cover the original binary masks. In addition, the organization of the images and annotations in the datasets was also adapted to the PASCAL VOC dataset format that our evaluation pipeline uses as input. This adaptation process usually required three steps: (*i*) folder reorganization, in which all original and mask images are moved into two separate folders if necessary; (*ii*) format conversion, to covert the original images to the JPG format if necessary; and (*iii*) conversion to PASCAL VOC dataset format, in which the dataset is adapted to this format, including the transformation of binary masks into bounding boxes when needed. 

The scripts to make such conversions were published in the following GitHub repository: https://github.com/sing-group/public-datasets-to-voc (accessed on 22 February 2022). The specific process of converting each of the datasets to this common format is discussed below. In Appendix A, we summarize the most relevant information regarding the datasets structure (number and image formats, scripts used to process them, etc.). This table also shows the number of bounding boxes obtained for each dataset along with the average relative bounding box size with respect to the whole image.

The CVC-ColonDB and CVC-ClinicDB datasets share the same folder structure and include several mask types for each image. The original polyp images are provided in their own folder, while the images of each mask type are placed in separate folders. All the images are provided in a BMP format, except for the “gtpolyp” mask images in CVC-ClinicDB, which are provided in a TIFF format. For our experiment, the “gtpolyp” mask images were used. The original polyp images were first converted to JPG using the convert_format.sh script. Finally, the dataset was converted into the PASCAL VOC dataset format using the CVC-ToVOC.py script.

The CVC-ClinicVideoDB dataset is structured as two folders, containing development (train and validation) and test partitions, and provides original polyp and mask images in a PNG format. Because the test partition does not include annotations, its images were discarded, and therefore only the images from the development partition were used. In this partition, original polyp and mask images are stored in separate folders by polyp. The original polyp and mask images were first separated into two different folders using the separate_folder_ClinicVideo.sh script, and then the original polyp images were converted into a JPG format using the convert_format.sh script. Finally, the dataset was converted into the PASCAL VOC dataset format using the ClinicVideoToVOC.py script.

The CVC-PolypHD dataset provides a single folder containing both the original polyp and mask images in BMP and TIFF formats, respectively. The original polyp and mask images were first separated into two different folders using the separate_folder_PolypHD.sh script, and then the original polyp images were converted into a JPG format using the convert_format.sh script. Finally, the dataset was converted into the PASCAL VOC dataset format using the PolypHDToVOC.py script.

The ETIS-Larib dataset is structured as two folders containing the original polyp and binary mask images in a TIFF format. The original polyp images were first converted into a JPG format using the convert_format.sh script, and then the dataset was converted into the PASCAL VOC dataset format using the ETIS-LaribToVOC.py script.

The Kvasir-SEG dataset is structured as two folders containing the original polyp and binary mask images in a JPG format and a JSON file that contains the bounding box locations of each image. The conversion of this dataset to the PASCAL VOC dataset format was carried out using the KvasirToVOC.py script.

The PICCOLO dataset is structured as three folders, containing the train, validation, and test partitions, and provides original polyp and mask images in PNG and TIFF formats, respectively. In this case, the original polyp and binary mask images were moved into two single separate folders using the merge_PICCOLO.sh script in order to get rid of the partitions and be able to use the whole dataset as a test set. Then, the original polyp images were converted into a JPG format using the convert_format.sh script, and the dataset was converted into the PASCAL VOC dataset format using the PICCOLOToVOC.py script.

The KUMC dataset is structured as three folders, containing the train, validation, and test partitions, respectively, which are already in the PASCAL VOC dataset format. In this case, we grouped the images into a single folder (i.e., merge train, validation, and test partitions) in order to be able to use the entire dataset as a test set. It is important to note that this dataset includes some annotations that do not have an image associated and, therefore, we excluded those annotations to create a usable version of this dataset. Also, this dataset includes labels for “adenomatous” and “hyperplastic” polyps, which were also merged into a single “polyp” annotation to be able to use them with our model (trained to locate images of class “polyp”). The whole conversion process of this dataset was carried out using the KUMCToVOC.sh script.

The SUN dataset provides one folder for each polyp, containing one or more images of the polyp in a JPG format and a text file with the bounding box location and the class (polyp vs. non-polyp) of each image. In this case, we grouped the images into a single folder in order to be able to use the entire dataset as a test set, using the merge_SUN.sh script. Finally, the dataset was converted into the PASCAL VOC dataset format using the SUNToVOC.sh script.

Finally, the LDPolypVideo dataset is structured as two folders, containing the development (train and validation) and test partitions, and provides original polyp images in a JPG format and a text file with the bounding box location of each image. In this case, we grouped the images into a single folder in order to be able to use the entire dataset as a test set, using the merge_and_rename_LDPolypVideo.sh script, which also renames the original image names to avoid duplicates when all images are put in the same folder. Finally, the dataset was converted into the PASCAL VOC dataset format using the LDPolypVideoToVOC.py script. 

### 2.4. Experiments

The experiments consisted in evaluating the performance of our model (presented in Section 2.1) in the ten public colonoscopy image datasets selected, without retrain. The model was developed using a set of Compi pipelines [32,33] available at this GitHub repository: https://github.com/sing-group/polydeep-object-detection (accessed on 22 February 2022). In order to carry out the experiments presented in this work, the test pipeline (test.xml) was used to load the trained model and analyze the performance on the ten public datasets after converting them to the PASCAL VOC format (as described in Section 2.3). This allowed us to obtain the performance results presented in Section 3.

### 2.5. Performance of Studies on Public Colonoscopy Image Datasets

Table 2 includes all published studies reporting bounding-box-based performance metrics (i.e., comparing predicted bounding boxes against the true bounding boxes of the ground truth) in at least one of the selected public colonoscopy image datasets, as resulted from the selection process explained in Section 2.2. These data were used then to analyze the performance of various models on the public datasets included in this study and compare our detection model with them. It is important to note that some studies evaluate the performance of several models and, in this case, we selected only the metrics of the best performing ones to perform our analyses and compare them.

The table includes one row for each study experiment with the following information: training set, testing set, recall, precision, F1-score, and F2-score. It is important to note that: (*i*) we only included the performance metrics for the selected public datasets, although some works reported performances with other datasets (e.g., Shin Y. et al., 2018 [34] reports the performance for the ASU-Mayo Clinic Colonoscopy Video, but it is not included, as we could not access the dataset); (*ii*) we included the performance metrics for private dataset partitions (Wang et al., 2018 [35], Wittenberg et al., 2019 [36], and Young Lee J. et al., 2020 [37]) since they able to compare those studies against us.

Each row of Table 2 (i.e., experiment performance) can be categorized as: (*i*) intra-dataset performance, when the evaluation was carried out on a test split of the dataset used for model development; or (*ii*) inter-dataset performance, when the performance evaluation was carried out on a dataset different than the one used for model development. From the 20 studies, there are 10 that only show their performance results for evaluating one public dataset, 8 that use at least two public datasets, 2 that use three datasets, and 1 that uses four public datasets.

**Table 2 diagnostics-12-00898-t002:** Performance results of studies evaluating DL models for polyp localization in at least one of the selected public colonoscopy image datasets.

Paper	Train	Test	Results			
			Recall	Precision	F1-Score	F2-Score
Brandao et al., 2018 [38]	CVC-ClinicDB + ASU-Mayo	ETIS-Larib	0.90	0.73	0.81	0.86
		CVC-ColonDB	0.90	0.80	0.85	0.88
Zheng Y. et al., 2018 [39]	CVC-ClinicDB + CVC-ColonDB	ETIS-Larib	0.74	0.77	0.76	0.75
Shin Y. et al., 2018 [34]	CVC-ClinicDB	ETIS-Larib	0.80	0.87	0.83	0.82
		CVC-Clinic VideoDB	0.84	0.90	0.87	0.85
Wang et al., 2018 [35]	Private	CVC-ClinicDB	0.88	0.93	0.91	0.89
		Private *	0.94	0.96	0.95	0.95
Qadir et al., 2019 [40]	CVC-ClinicDB	CVC-ClinicVideoDB	0.84	0.90	0.87	0.85
Tian Y. et al., 2019 [41]	Private	ETIS-Larib	0.64	0.74	0.69	0.66
Ahmad et al., 2019 [42]	Private	ETIS-Larib	0.92	0.75	0.83	0.88
Sornapudi et al., 2019 [43]	CVC-ClinicDB	ETIS-Larib	0.80	0.73	0.76	0.79
		CVC-ColonDB	0.92	0.90	0.91	0.91
		CVC-PolypHD	0.78	0.83	0.81	0.79
Wittenberg et al., 2019 [36]	Private	ETIS-Larib	0.83	0.74	0.79	0.81
		CVC-ClinicDB	0.86	0.80	0.82	0.85
		Private	0.93	0.86	0.89	0.92
Jia X. et al., 2020 [44]	CVC-ColonDB	CVC-ClinicDB	0.92	0.85	0.88	0.91
	CVC-ClinicDB	ETIS-Larib	0.82	0.64	0.72	0.77
Ma Y. et al., 2020 [45]	CVC-ClinicDB	CVC-ClinicVideoDB	0.92	0.88	0.90	0.91
Young Lee J. et al., 2020 [37]	Private	CVC-ClinicDB	0.90	0.98	0.94	0.96
		Private	0.97	0.97	0.97	0.97
Podlasek J. et al., 2020 [46]	Private	ETIS-Larib	0.67	0.79	0.73	0.69
		CVC-ClinicDB	0.91	0.97	0.94	0.92
		CVC-ColonDB	0.74	0.92	0.82	0.77
		Hyper-Kvasir	0.88	0.98	0.93	0.90
Qadir et al., 2021 [47]	CVC-ClinicDB	ETIS-Larib	0.87	0.86	0.86	0.86
		CVC-ColonDB	0.91	0.88	0.90	0.90
Xu J. et al., 2021 [48]	CVC-ClinicDB	ETIS-Larib	0.72	0.83	0.77	0.74
		CVC-ClinicVideoDB	0.66	0.89	0.76	0.70
Pacal et al., 2021 [49]	CVC-ClinicDB	ETIS-Larib	0.83	0.92	0.87	0.84
		CVC-ColonDB	0.97	0.96	0.96	0.97
Liu et al., 2021 [50]	CVC-ClinicDB	ETIS-Larib	0.88	0.78	0.82	0.85
Li K. et al., 2021 [23]	KUMC	KUMC-Test **	0.86	0.91	0.89	0.87
Ma Y. et al., 2021 [21]	CVC-ClinicDB	CVC-ClinicVideoDB	0.64	0.85	0.73	0.67
		LDPolypVideo	0.47	0.65	0.55	0.50
Pacal et al., 2022 [51]	SUN + PICCOLO + CVC-ClinicDB	ETIS-Larib	0.91	0.91	0.91	0.91
	SUN	SUN ***	0.86	0.96	0.91	0.88
	PICCOLO	PICCOLO	0.80	0.93	0.86	0.82

* Wang et al., 2018 evaluated the test performance using a different private dataset from the one used for model training. However, we consider this as an intra-dataset experiment since the private dataset for model development was collected in the Endoscopy Center of Sichuan Provincial People’s Hospital between January 2007 and December 2015 and the private test dataset was collected in the same center using the same devices between January and December 2016, and we understand that the distribution should be very similar. ** Li K. et al., 2021 used a partition of the KUMC dataset as testing set in their experiments (KUMC-Test in the table). *** Pacal et al., 2022 used a partition of the SUN dataset that includes “non-polyp” images and, therefore, it is not comparable to our performance with the SUN dataset, which includes all polyp images.

Among the public datasets, as Figure 4 shows, the ETIS-Larib dataset was the most widely used for testing the detection models (14 out of 20 studies), probably due to the fact that it was one of the test datasets for the automatic polyp detection subchallenge at MICCAI 2015 [11]. The highest F1-score in this dataset was achieved by Pacal et al., 2022 [51] (0.91). The next datasets used by the greatest number of studies (5 out of 20) were CVC-ColonDB, for which the highest F1 (0.96) was achieved by Pacal et al., 2021 [49], CVC-ClinicDB, for which both Young Lee J. et al., 2020 [37] and Podlasek J. et al., 2020 [46] achieved an F1 of 0.94, and CVC-ClinicVideoDB, for which Ma Y. et al., 2020 [45] achieved the top F1-score of 0.90.

## 3. Results and Discussion

Table 3 shows the performance results of our model when evaluated on the ten selected public colonoscopy image datasets. As shown in Figure 5, the F1-score of our model decayed in all public datasets with respect to the performance in our private test partition (F1 = 0.88, recall = 0.87, precision = 0.89). The average F1 decay was 13.65%, reaching its maximum with the whole LDPolypVideo dataset (F1 = 0.52), for which we also had the lowest recall (0.49). 

The three datasets in which our model decayed the most were LDPolypVideo (−40.75%), PICCOLO (−24.27%), and ETIS-Larib (−18.54%). These datasets share two characteristics that the private dataset used to develop and test our model does not have: (*i*) the presence of non-polyp images (LDPolypVideo and PICCOLO), which may decrease our precision as we are showing more test images without polyps and our model has more chances to emit false positives; and (*ii*) the presence of images with multiple-polyp images (LDPolypVideo, PICCOLO, and ETIS-Larib), which may decrease our recall even though our model is able to locate multiple polyps (e.g., PICCOLO contains almost 10% of images annotated with multiple polyps, and our recall in this dataset (0.60) was significantly lower than in others).

The low performance in the LDPolypVideo dataset is not surprising, as authors state in their publication that the dataset contains images selected to include a high degree of diversity in polyp morphology, multiple polyps, motion blur, and specular reflections, in order to create a challenging dataset [21]. In fact, they fine-tuned several state-of-the-art object detection models (including YOLOv3, the same as us) using the CVC-ClinicDB dataset and evaluated their performance using the CVC-ClinicVideoDB dataset and their new LDPolypVideo dataset, obtaining a significantly lower performance in the LDPolypVideo dataset evaluation with all models. Their best F1-score (0.55) in the LDPolypVideo dataset was obtained using RetinaNet, while their YOLOv3 model obtained an F1-score of 0.41 (compared to our F1-score of 0.52).

Intrigued by the low F1-score in the whole PICCOLO dataset, we analyzed the performance of our model in the three original partitions of the dataset separately, obtaining an F1-score 0.71 in the train partition (recall = 0.63, precision = 0.80, 2203 images), an F1-score 0.53 in the validation partition (recall = 0.49, precision = 0.61, 897 images), and an F1-score 0.74 in the test partition (recall = 0.69, precision = 0.80, 333 images). As can be seen, the performance in the validation partition was significantly worse than in the other two partitions, which we believe to be the main cause of the performance decrease when testing with the whole dataset. Figure 6 shows several incorrect predictions of our model against the ground truth in the train, validation, and tests splits of the PICCOLO dataset. We noted that the validation split contains many big bounding boxes as ground truth, and we computed the average relative size of the bounding boxes with respect to the whole image in the three partitions, obtaining 0.20 in train, 0.33 in validation, and 0.16 in test. Thus, bounding boxes in the validation set are clearly bigger than in the other two partitions. We also observed the majority of polyps in the validation partition look like the three images shown in the middle column of Figure 6, while polyps in our dataset look like the three images taken from the PICCOLO training set in the first column. We understand that polyp images in the validation set, where our model decayed the most, follow a different distribution than the ones in our private training set. Nevertheless, some of these errors (seen in the bottom-left or upper-right images in Figure 6) are caused because the intersection between the predicted and the actual bounding boxes is below the threshold, but in practical terms, an endoscopist would be able to localize the polyp in real-time when using the model.

Only 3 out of the 20 studies had the same exact setup that we had: training the model with a private dataset, testing it with a test partition of the private dataset (intra-dataset performance), and finally testing it using one or more public datasets (inter-dataset performance estimation). As shown in Figure 7, the F1-score also decayed in those studies when analyzing the public datasets with respect to the private test set. In the case of the CVC-ClinicDB dataset, used by the four studies, the average decay was about 5%. In the ETIS-Larib dataset, the Wittenberg et al., 2019 [36] study decayed by 11.24%, compared to our 18.54% decrease.

With the aim of further exploring the intra-to-inter performance decay in other studies, we analyzed the evaluations on public datasets collected in Section 2.5. Such evaluations are heterogeneous regarding the datasets used for training and testing, the models used, and other similar factors. Thus, we compared the intra-dataset performances (i.e., a performance evaluation on a test split of the dataset used for model development, either private or public) against the inter-dataset performances (i.e., a performance evaluation on a dataset different than the one used for model development). As Figure 8A shows, the intra-dataset performances were usually higher than the inter-dataset performances. Figure 8B shows the inter-dataset distribution disaggregated by the public dataset on which the evaluation was carried out. It is important to note that three public datasets (PICCOLO, KUMC, and SUN) are under the intra-dataset performance box, as they are only used in intra-dataset setups. These results are in line with the results obtained in our experiment that show a decay in the F1-score when the evaluation was carried out on a different test dataset.

Interestingly, the two datasets in which the decay of our model was lower (CVC-ClinicDB and CVC-ColonDB) are also two of the three datasets in which the published studies obtained performances closer to the intra-dataset ones; also, two of the three datasets in which the performance was worse in the published studies (ETIS-Larib and LDPolypVideo) were two of the most challenging for our model. In the light of these observations, we correlated our performance in the seven datasets shown in Figure 8B with the median inter-dataset performances of the published studies. Figure 9 allows us to observe that such correlation exists (*p*-value < 0.001), showing that our exhaustive testing reveals the inherent degree of difficulty of the public datasets; a single model (ours) showed the same behavior as the aggregation of the published research, taking into account that studies are heterogeneous (different models and different training datasets) and that we picked the highest F1-scores of those performing several analyses.

## 4. Conclusions

In this work, we performed the biggest systematic evaluation of a polyp localization model trained using a private dataset and tested it on ten public colonoscopy image datasets, including the most recent PICCOLO, SUN, and KUMC datasets. The biggest evaluation to date was carried out by Podlasek J. et al., 2020 [46], who tested their model with four datasets. As a result of performing such an evaluation, we have published a set of scripts for converting the public datasets into the PASCAL VOC format for polyp localization, providing a valuable resource for other researchers aiming to perform similar analyses.

Our experiments and the analysis of the published research allowed us to observe that there is a performance decay when performing an inter-dataset evaluation. The F1-score of our model was 0.88 when evaluated on a private test partition and decayed, on average, 13.65% when tested on the ten public datasets selected. In the published research, the average F1-score was 0.91 when the evaluation was performed on a test split of the dataset used for mode development, compared to the 0.83 average F1 obtained when such models were tested with external datasets, keeping in mind that these F1-scores are the best ones among the reported performances. This confirms our initial hypothesis that models developed using one dataset are sensitive to changes in the nature of the testing instances. Also, we observed that this decay is associated with the test dataset; while studies on datasets such as CVC-ClinicDB and CVC-ColonDB obtain F1-scores closer to their development performances, other datasets such as ETIS-Larib and LDPolypVideo, are more challenging.

In light of these findings, our future work to keep improving our model will be data-centric. In the first place, we will use an updated version of our dataset (now available through the IISGS BioBank: https://www.iisgaliciasur.es/home/biobanco/cohorte-pibadb (accessed on 22 February 2022)) that includes more annotated images for polyp localization. In this updated version, we have also included annotated non-polyp images, which some studies also used during training to improve the performance of the model [22]. Finally, we will also evaluate the possibility of augmenting our training data with public datasets, as some studies have attempted [51], giving priority to those datasets where our model decays the most, such as ETIS-Larib, PICCOLO, or LDPolypVideo. Doing this would also allow us to train our model using images annotated with multiple polyps.

## Figures and Tables

**Figure 1 diagnostics-12-00898-f001:**
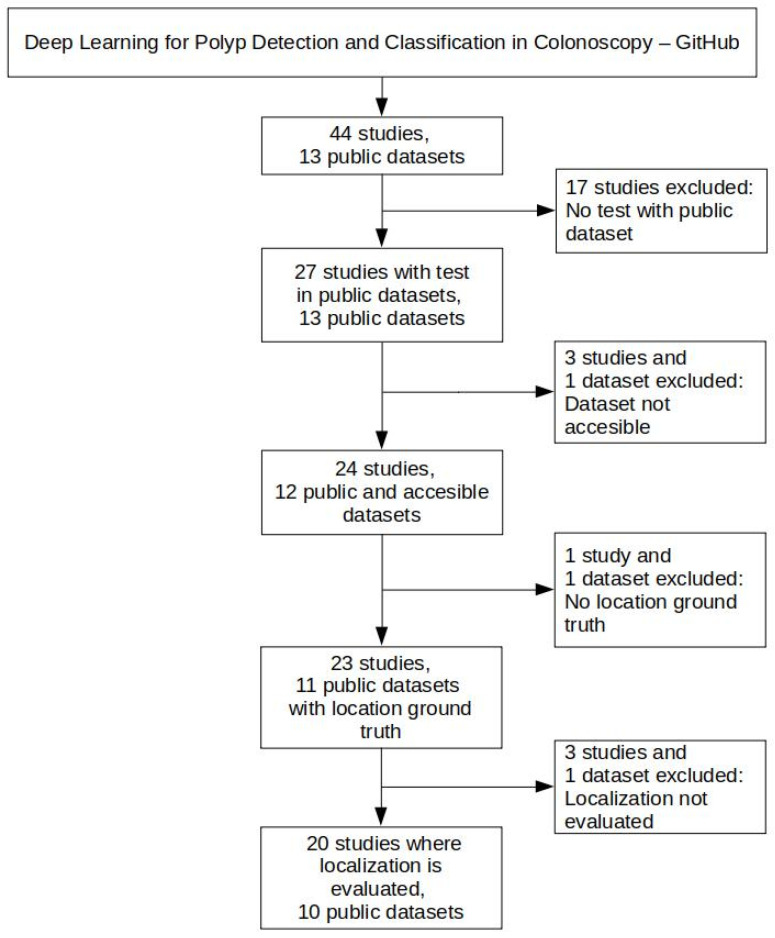
Criteria for selecting polyp localization studies and public colonoscopy image datasets.

**Figure 2 diagnostics-12-00898-f002:**
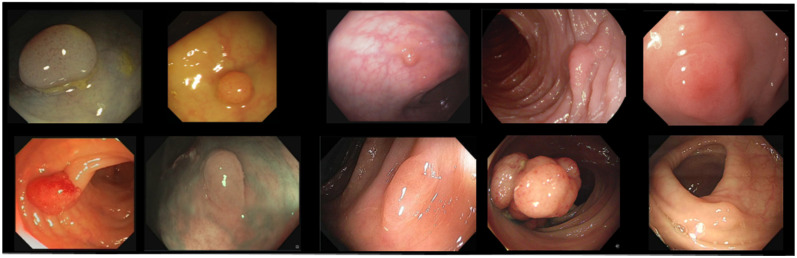
Examples of polyp images from the included datasets. Upper row (left to right): CVC-ClinicDB, CVC-ColonDB, CVC-PolypHD, ETIS-Larib, and Kvasir-SEG. Bottom row (left to right): CVC-ClinicVideoDB, PICCOLO, KUMC, SUN, and LDPolypVideo.

**Figure 3 diagnostics-12-00898-f003:**
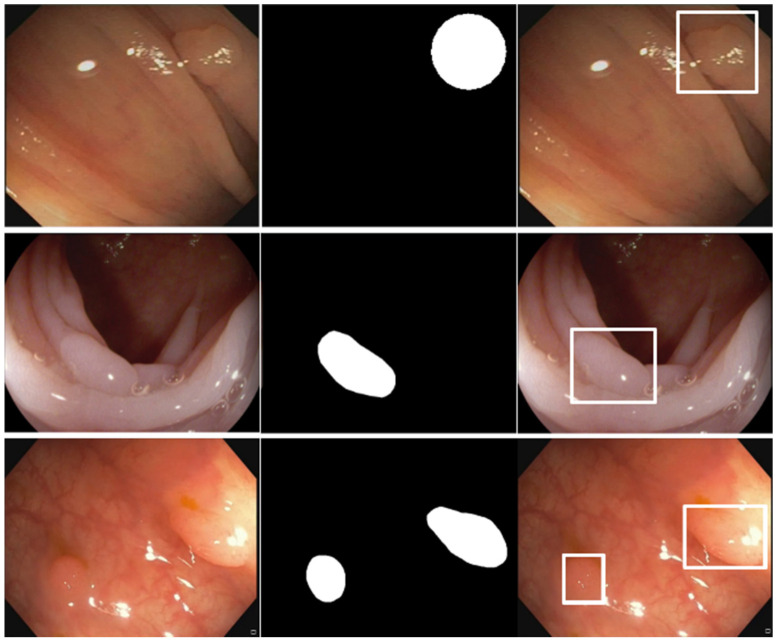
Conversion from binary mask annotations to bounding boxes. First column: original polyp images. Second column: binary mask annotations. Third column: obtained bounding box annotations over the original polyp images.

**Figure 4 diagnostics-12-00898-f004:**
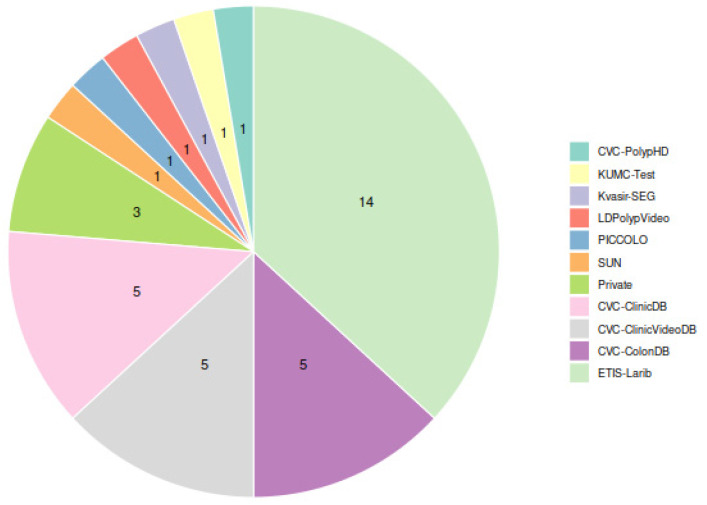
Usage of datasets for model evaluation among studies in Table 2. Each study using several datasets contributes one point for each testing dataset used.

**Figure 5 diagnostics-12-00898-f005:**
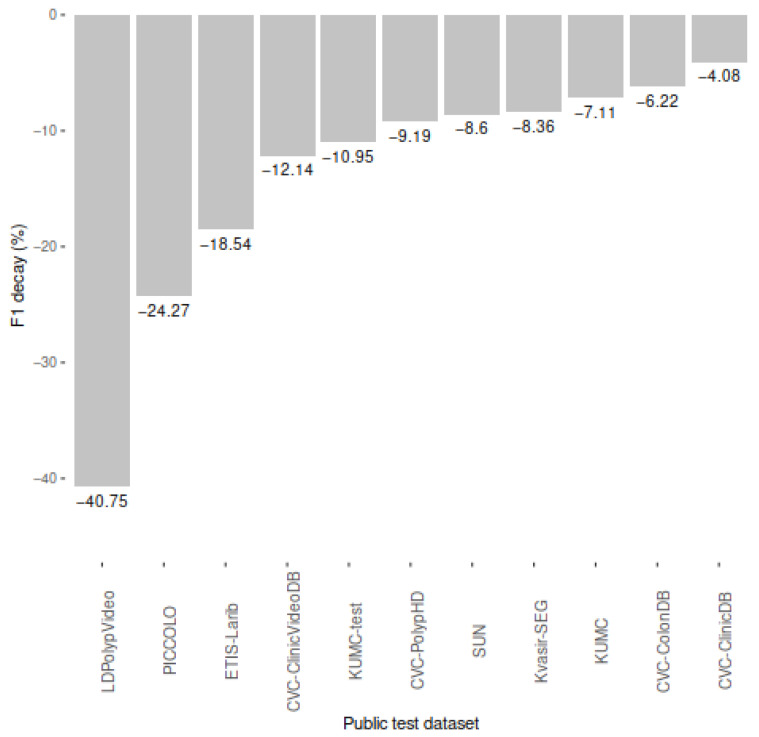
F1 decay (%) in public colonoscopy image datasets compared to the performance of our model in our private dataset reported in Nogueira-Rodríguez et al., 2021 (0.88) [26].

**Figure 6 diagnostics-12-00898-f006:**
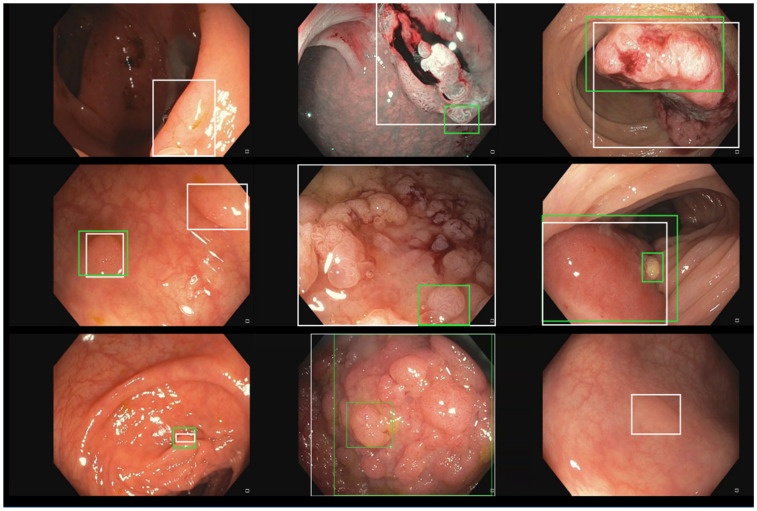
Incorrect prediction examples of our detection model over the different splits of the PICCOLO dataset. From left to right, there are three examples taken from the train, validation, and test splits. Predicted boxes are depicted in green, whereas ground truth boxes are in white.

**Figure 7 diagnostics-12-00898-f007:**
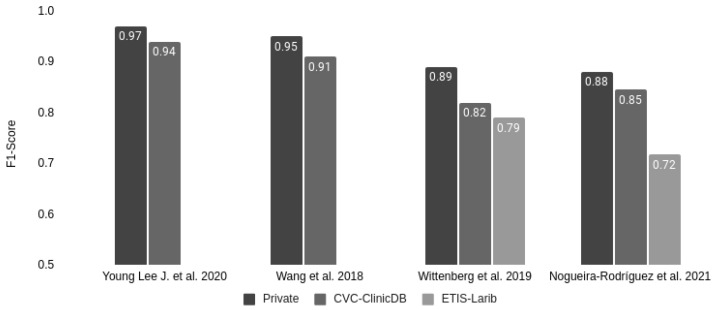
Comparison of F1-score decay on public colonoscopy image datasets of those studies reporting their performance for the private test set partition.

**Figure 8 diagnostics-12-00898-f008:**
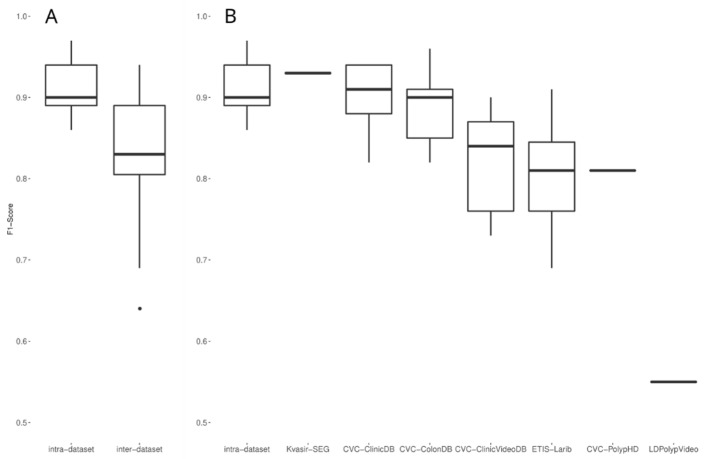
(**A**) Comparison of intra-dataset and inter-dataset performances of the 20 selected studies. (**B**) Same as A, with the inter-dataset performances disaggregated by dataset.

**Figure 9 diagnostics-12-00898-f009:**
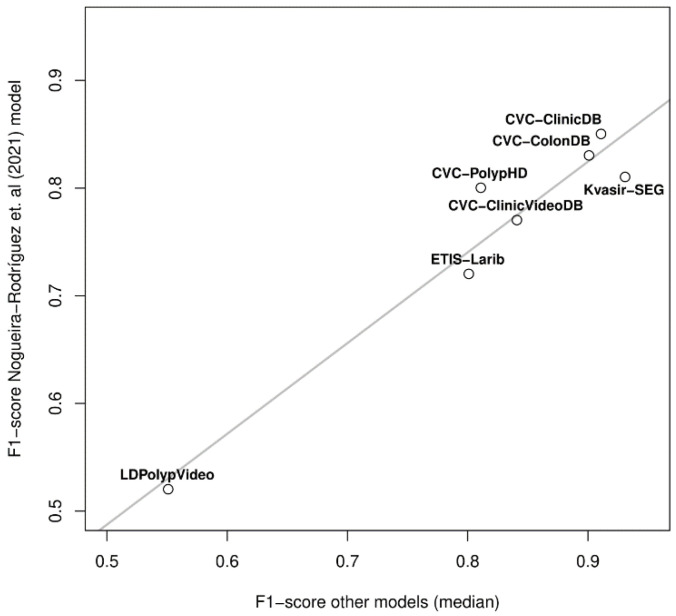
Correlation between the median inter-dataset F1-score of published studies and ours in seven public colonoscopy image datasets.

**Table 1 diagnostics-12-00898-t001:** Descriptions of the ten public colonoscopy image datasets for polyp localization.

Dataset	Paper Publication Year	Description	Resolution	Ground Truth	Presence of Multiple Polyp Images	Presence of Non-Polyp Images
CVC-ClinicDB [12]	2015	612 sequential WL images with polyps extracted from 31 sequences (23 patients) with 31 different polyps	384 × 288	Binary mask to locate the polyp	yes	no
CVC-ColonDB [15,16]	2012	300 sequential WL images with polyps extracted from 13 sequences (13 patients)	574 × 500	Binary mask to locate the polyp	no	no
CVC-PolypHD [15,16]	2018	56 WL images	1920 × 1080	Binary mask to locate the polyp	yes	no
ETIS-Larib [13]	2014	196 WL images with polyps extracted from 34 sequences with 44 different polyps	1225 × 966	Binary mask to locate the polyp	yes	no
Kvasir-SEG [20]	2020	1000 polyp images	332 × 487 1920 × 1072	Binary mask and bounding box to locate the polyp	yes	no
CVC-ClinicVideoDB [17,18]	2017	11,954 images in total with 10,025 images of polyps	384 × 288	Binary mask to locate the polyp	no	yes
PICCOLO [19]	2020	3433 images (2131 WL and 1302 NBI) from 76 lesions from 40 patients	854 × 480 1920 × 1080	Binary mask to locate the polyp	yes	yes
KUMC dataset [23]	2021	37,899 images in total, including the CVC-ColonDB, ASU-Mayo Clinic Colonoscopy Video, and Colonoscopic Dataset datasets	Various resolutions	Bounding box to locate the polyp	no	yes
SUN [22]	2021	49,136 images with polyps. The polyp samples of 100 cases	1240 × 1080	Bounding box to locate the polyp	no	no *
LDPolypVideo [21]	2021	160 videos (40,187 frames: 33,876 polyp images and 6311 non-polyp images) with 200 labeled polyps.	560 × 480	Bounding box to locate the polyp	yes	yes

* The SUN dataset contains 109,554 non-polyp frames that were not downloaded for our experiments.

**Table 3 diagnostics-12-00898-t003:** Performance results of our model when evaluated on the ten selected public colonoscopy image datasets.

Dataset	Number of Images for Test	Results				
		Recall	Precision	F1-Score	F2-Score	AP
CVC-ClinicDB	612	0.82	0.87	0.85	0.83	0.82
CVC-ColonDB	300	0.84	0.81	0.83	0.83	0.85
CVC-PolypHD	56	0.75	0.86	0.80	0.77	0.79
ETIS-Larib	196	0.72	0.71	0.72	0.72	0.69
Kvasir-SEG	1000	0.78	0.84	0.81	0.82	0.79
PICCOLO	3433	0.60	0.76	0.67	0.62	0.63
CVC-ClinicVideoDB	11,954	0.80	0.75	0.77	0.79	0.77
KUMC dataset	37,899	0.81	0.83	0.82	0.81	0.83
KUMC dataset–Test	4872	0.76	0.81	0.78	0.77	0.79
SUN	49,136	0.78	0.83	0.81	0.79	0.81
LDPolypVideo	40,186	0.49	0.56	0.52	0.50	0.44

## Data Availability

Publicly available datasets were analyzed in this study. The CVC-ClinicDB, CVC-ColonDB, CVC-ClinicVideoDB, and CVC-PolypHD datasets are publicly available here: https://giana.grand-challenge.org. The ETIS-Larib dataset is publicly available here: https://polyp.grand-challenge.org/EtisLarib. The Kvasir-SEG dataset is publicly available here: https://datasets.simula.no/kvasir-seg. The PICCOLO dataset is publicly available here: https://www.biobancovasco.org/en/Sample-and-data-catalog/Databases/PD178-PICCOLO-EN.html. The KUMC dataset is publicly available here: https://dataverse.harvard.edu/dataset.xhtml?persistentId=doi:10.7910/DVN/FCBUOR. The SUN dataset is publicly available here: http://amed8k.sundatabase.org/. The LDPolypVideo dataset is publicly available here: https://github.com/dashishi/LDPolypVideo-Benchmark (accessed on 22 February 2022).

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
