# Peer review of "Performance of Convolutional Neural Networks for Polyp Localization on Public Colonoscopy Image Datasets"

_diagnostics, 2022, doi:10.3390/diagnostics12040898_

Round 1

Reviewer 1 Report

I really appreciate the effort by the authors for their contribution and especially for the GitHub page, where the readers can find papers for polyp localization and detection. 

https://github.com/sing-group/deep-learning-colonoscopy

  1. The paper is good in terms of the survey. However, I find the technical novelty limited. 

  2. The experiments on 10 datasets are highly appreciable. I also appreciate that the authors first converted the bounding box into the VOC format for their experiments. However, from the research point of view, I would like to see how other methods performed on the same train-test split.

  3. Please include operational efficiency that is critical to the implementation of algorithms.

  4. Please include speed that is critical for clinical settings. 

  5. I have been following the GitHub page for a year. As the CVC series, datasets are old. The authors seem to report more work on CVC-series datasets. The authors should also conclude the works on the recent datasets that are done on public datasets. 

Jha, D., Ali, S., Tomar, N. K., Johansen, H. D., Johansen, D., Rittscher, J., ... & Halvorsen, P. (2021). Real-time polyp detection, localization and segmentation in colonoscopy using deep learning. Ieee Access, 9, 40496-40510.

Pacal, I., & Karaboga, D. (2021). A robust real-time deep learning based automatic polyp detection system. Computers in Biology and Medicine, 134, 104519.

Li, K., Fathan, M. I., Patel, K., Zhang, T., Zhong, C., Bansal, A., ... & Wang, G. (2021). Colonoscopy polyp detection and classification: Dataset creation and comparative evaluations. Plos one, 16(8), e0255809.

  1. It is suggested to include 2021 challenges and competitions on polyp detection and localization to make your work more comprehensive. 

https://endocv2021.grand-challenge.org/

Challenge overview paper:

Ali, S., Ghatwary, N., Jha, D., Isik-Polat, E., Polat, G., Yang, C., ... & East, J. E. (2022). Assessing generalisability of deep learning-based polyp detection and segmentation methods through a computer vision challenge. arXiv preprint arXiv:2202.12031.

Dataset: (PolypGen)- Description can be found in the paper. 

Ali, S., Jha, D., Ghatwary, N., Realdon, S., Cannizzaro, R., Salem, O. E., ... & East, J. E. (2021). PolypGen: A multi-center polyp detection and segmentation dataset for generalisability assessment. arXiv preprint arXiv:2106.04463.

Some of the works on detection can be found here:

http://ceur-ws.org/Vol-2886/

  1. In Table 3, please cite the dataset. 

  2. Please also check the citations for all the datasets in the Table. I found the citations for Kvasir-SEG incorrect. Please check and correct others. 

Reviewer 2 Report

The authors reported the results of tests of their previously published polyp detection model by using ten public colonoscopy image datasets and analyzed them in the context of results from other state-of-the-art publications using the same datasets. This study broadly validates the usability of the model, which has some utility, and is worthy of publication. 

Author Response

We thank the reviewer for his/her appreciation of our work.

Round 2

Reviewer 1 Report

The overall quality of the work is good. 

I would recommend the authors incorporate the following efforts made by authors in their paper. They are doing similar works like yours. 

https://github.com/GewelsJI/VPS/blob/main/docs/AWESOME_VPS.md

Additionally, it is understandable that making changes to one Tables might affect others. Therefore, I would recommend the manuscript for publication without further changes.